# $k^*$-Nearest Neighbors: From Global to Local

**Oren Anava**
The Voleon Group
oren@voleon.com

**Kfir Y. Levy**
ETH Zurich
yehuda.levy@inf.ethz.ch

## Abstract

The weighted $k$-nearest neighbors algorithm is one of the most fundamental non-parametric methods in pattern recognition and machine learning. The question of setting the optimal number of neighbors as well as the optimal weights has received much attention throughout the years, nevertheless this problem seems to have remained unsettled. In this paper we offer a simple approach to locally weighted regression/classification, where we make the bias-variance tradeoff explicit. Our formulation enables us to phrase a notion of optimal weights, and to efficiently find these weights as well as the optimal number of neighbors *efficiently and adaptively, for each data point whose value we wish to estimate*. The applicability of our approach is demonstrated on several datasets, showing superior performance over standard locally weighted methods.

## 1 Introduction

The $k$-nearest neighbors ($k$-NN) algorithm [1, 2], and Nadarays-Watson estimation [3, 4] are the cornerstones of non-parametric learning. Owing to their simplicity and flexibility, these procedures had become the methods of choice in many scenarios [5], especially in settings where the underlying model is complex. Modern applications of the $k$-NN algorithm include recommendation systems [6], text categorization [7], heart disease classification [8], and financial market prediction [9], amongst others.

A successful application of the weighted $k$-NN algorithm requires a careful choice of three ingredients: the number of nearest neighbors $k$, the weight vector $\boldsymbol{\alpha}$, and the distance metric. The latter requires domain knowledge and is thus henceforth assumed to be set and known in advance to the learner. Surprisingly, even under this assumption, the problem of choosing the optimal $k$ and $\boldsymbol{\alpha}$ is not fully understood and has been studied extensively since the 1950's under many different regimes. Most of the theoretic work focuses on the asymptotic regime in which the number of samples $n$ goes to infinity [10, 11, 12], and ignores the practical regime in which $n$ is finite. More importantly, the vast majority of $k$-NN studies aim at finding an optimal value of $k$ per dataset, which seems to overlook the specific structure of the dataset and the properties of the data points whose labels we wish to estimate. While kernel based methods such as Nadaraya-Watson enable an adaptive choice of the weight vector $\boldsymbol{\alpha}$, theres still remains the question of how to choose the *kernel's bandwidth $\sigma$*, which could be thought of as the parallel of the number of neighbors $k$ in $k$-NN. Moreover, there is no principled approach towards choosing the kernel function in practice.

In this paper we offer a coherent and principled approach to *adaptively* choosing the number of neighbors $k$ and the corresponding weight vector $\boldsymbol{\alpha} \in \mathbb{R}^k$ per decision point. Given a new decision point, we aim to find the best locally weighted predictor, in the sense of minimizing the distance between our prediction and the ground truth. In addition to yielding predictions, our approach enables us to provide a *per decision point* guarantee for the confidence of our predictions. Fig. 1 illustrates the importance of choosing $k$ adaptively. In contrast to previous works on non-parametric regression/classification, we do not assume that the data $\{(x_i, y_i)\}_{i=1}^n$ arrives from some (unknown) underlying distribution, but rather make a weaker assumption that the labels $\{y_i\}_{i=1}^n$ are independent

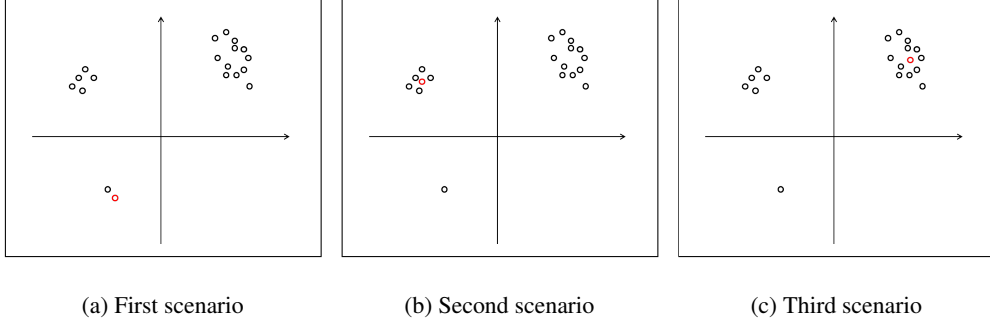

|              (a) First scenario              |              (b) Second scenario              |              (c) Third scenario              |

Figure 1: Three different scenarios. In all three scenarios, the same data points $x_1, \ldots, x_n \in \mathbb{R}^2$ are given (represented by black dots). The red dot in each of the scenarios represents the new data point whose value we need to estimate. Intuitively, in the first scenario it would be beneficial to consider only the nearest neighbor for the estimation task, whereas in the other two scenarios we might profit by considering more neighbors.

given the data points $\{x_i\}_{i=1}^n$, allowing the latter to be chosen arbitrarily. Alongside providing a theoretical basis for our approach, we conduct an empirical study that demonstrates its superiority with respect to the state-of-the-art.

This paper is organized as follows. In Section 2 we introduce our setting and assumptions, and derive the locally optimal prediction problem. In Section 3 we analyze the solution of the above prediction problem, and introduce a greedy algorithm designed to *efficiently* find the *exact* solution. Section 4 presents our experimental study, and Section 5 concludes.

## 1.1 Related Work

Asymptotic universal consistency is the most widely known theoretical guarantee for $k$-NN. This powerful guarantee implies that as the number of samples $n$ goes to infinity, and also $k \to \infty$, $k/n \to 0$, then the risk of the $k$-NN rule converges to the risk of the Bayes classifier for any underlying data distribution. Similar guarantees hold for weighted $k$-NN rules, with the additional assumptions that $\sum_{i=1}^k \alpha_i = 1$ and $\max_{i \leq n} \alpha_i \to 0$, [12, 10]. In the regime of practical interest where the number of samples $n$ is finite, using $k = \lfloor \sqrt{n} \rfloor$ neighbors is a widely mentioned rule of thumb [10]. Nevertheless, this rule often yields poor results, and in the regime of finite samples it is usually advised to choose $k$ using cross-validation. Similar consistency results apply to kernel based local methods [13, 14].

A novel study of $k$-NN by Samworth, [11], derives a closed form expression for the optimal weight vector, and extracts the optimal number of neighbors. However, this result is only optimal under several restrictive assumptions, and only holds for the asymptotic regime where $n \to \infty$. Furthermore, the above optimal number of neighbors/weights do not adapt, but are rather fixed over all decision points given the dataset. In the context of kernel based methods, it is possible to extract an expression for the optimal kernel's bandwidth $\sigma$ [14, 15]. Nevertheless, this bandwidth is fixed over all decision points, and is only optimal under several restrictive assumptions.

There exist several heuristics to adaptively choosing the number of neighbors and weights separately for each decision point. In [16, 17] it is suggested to use local cross-validation in order to adapt the value of $k$ to different decision points. Conversely, Ghosh [18] takes a Bayesian approach towards choosing $k$ adaptively. Focusing on the multiclass classification setup, it is suggested in [19] to consider different values of $k$ for each class, choosing $k$ proportionally to the class populations. Similarly, there exist several attitudes towards adaptively choosing the kernel's bandwidth $\sigma$, for kernel based methods [20, 21, 22, 23].

Learning the distance metric for $k$-NN was extensively studied throughout the last decade. There are several approaches towards metric learning, which roughly divide into linear/non-linear learning methods. It was found that metric learning may significantly affect the performance of $k$-NN in numerous applications, including computer vision, text analysis, program analysis and more. A comprehensive survey by Kulis [24] provides a review of the metric learning literature. Throughout

this work we assume that the distance metric is fixed, and thus the focus is on finding the best (in a sense) values of $k$ and $\boldsymbol{\alpha}$ for each new data point.

Two comprehensive monographs, [10] and [25], provide an extensive survey of the existing literature regarding $k$-NN rules, including theoretical guarantees, useful practices, limitations and more.

## 2  Problem Definition

In this section we present our setting and assumptions, and formulate the locally weighted optimal estimation problem. Recall we seek to find the best local prediction in a sense of minimizing the distance between this prediction and the ground truth. The problem at hand is thus defined as follows: We are given $n$ data points $x_1, \ldots, x_n \in \mathbb{R}^d$, and $n$ corresponding labels[1] $y_1, \ldots, y_n \in \mathbb{R}$. Assume that for any $i \in \{1, \ldots, n\} = [n]$ it holds that $y_i = f(x_i) + \epsilon_i$, where $f(\cdot)$ and $\epsilon_i$ are such that:

(1) **f$(\cdot)$ is a Lipschitz continuous function:** For any $x, y \in \mathbb{R}^d$ it holds that $|f(x) - f(y)| \leq L \cdot d(x, y)$, where the distance function $d(\cdot, \cdot)$ is set and known in advance. This assumption is rather standard when considering nearest neighbors-based algorithms, and is required in our analysis to bound the so-called *bias* term (to be later defined). In the *binary classification* setup we assume that $f : \mathbb{R}^d \mapsto [0, 1]$, and that given $x$ its label $y \in \{0, 1\}$ is distributed Bernoulli$(f(x))$.

(2) **$\epsilon_i$'s are noise terms:** For any $i \in [n]$ it holds that $\mathbb{E}\left[\epsilon_i | x_i\right] = 0$ and $|\epsilon_i| \leq b$ for some given $b > 0$. In addition, it is assumed that given the data points $\{x_i\}_{i=1}^n$ then the noise terms $\{\epsilon_i\}_{i=1}^n$ are independent. This assumption is later used in our analysis to apply Hoeffding's inequality and bound the so-called *variance* term (to be later defined). Alternatively, we could assume that $\mathbb{E}\left[\epsilon_i^2 | x_i\right] \leq b$ (instead of $|\epsilon_i| \leq b$), and apply Bernstein inequalities. The results and analysis remain qualitatively similar.

Given a new data point $x_0$, our task is to estimate $f(x_0)$, where we restrict the estimator $\hat{f}(x_0)$ to be of the form $\hat{f}(x_0) = \sum_{i=1}^n \alpha_i y_i$. That is, the estimator is a weighted average of the given noisy labels. Formally, we aim at minimizing the absolute distance between our prediction and the ground truth $f(x_0)$, which translates into

$$\min_{\boldsymbol{\alpha} \in \Delta_n} \left| \sum_{i=1}^n \alpha_i y_i - f(x_0) \right| \qquad (\textbf{P1}),$$

where we minimize over the simplex, $\Delta_n = \{\boldsymbol{\alpha} \in \mathbb{R}^n | \sum_{i=1}^n \alpha_i = 1 \text{ and } \alpha_i \geq 0, \ \forall i\}$. Decomposing the objective of ($\textbf{P1}$) into a sum of bias and variance terms, we arrive at the following relaxed objective:

$$\left| \sum_{i=1}^n \alpha_i y_i - f(x_0) \right| = \left| \sum_{i=1}^n \alpha_i \left( y_i - f(x_i) + f(x_i) \right) - f(x_0) \right|$$

$$= \left| \sum_{i=1}^n \alpha_i \epsilon_i + \sum_{i=1}^n \alpha_i \left( f(x_i) - f(x_0) \right) \right|$$

$$\leq \left| \sum_{i=1}^n \alpha_i \epsilon_i \right| + \left| \sum_{i=1}^n \alpha_i \left( f(x_i) - f(x_0) \right) \right|$$

$$\leq \left| \sum_{i=1}^n \alpha_i \epsilon_i \right| + L \sum_{i=1}^n \alpha_i d(x_i, x_0).$$

By Hoeffding's inequality (see supplementary material) it follows that $|\sum_{i=1}^n \alpha_i \epsilon_i| \leq C\|\boldsymbol{\alpha}\|_2$ for $C = b\sqrt{2 \log \left( \frac{2}{\delta} \right)}$, w.p. at least $1 - \delta$. We thus arrive at a new optimization problem ($\textbf{P2}$), such that solving it would yield a guarantee for ($\textbf{P1}$) with high probability:

$$\min_{\boldsymbol{\alpha} \in \Delta_n} C\|\boldsymbol{\alpha}\|_2 + L \sum_{i=1}^n \alpha_i d(x_i, x_0) \qquad (\textbf{P2}).$$

The first term in (**P2**) corresponds to the noise in the labels and is therefore denoted as the *variance* term, whereas the second term corresponds to the distance between $f(x_0)$ and $\{f(x_i)\}_{i=1}^n$ and is thus denoted as the *bias* term.

## 3 Algorithm and Analysis

In this section we discuss the properties of the optimal solution for (**P2**), and present a greedy algorithm which is designed in order to efficiently find the exact solution of the latter objective (see Section 3.1). Given a decision point $x_0$, Theorem 3.1 demonstrates that the optimal weight $\alpha_i$ of the data point $x_i$ is proportional to $-d(x_i, x_0)$ (closer points are given more weight). Interestingly, this weight decay is quite slow compared to popular weight kernels, which utilize sharper decay schemes, e.g., exponential/inversely-proportional. Theorem 3.1 also implies a cutoff effect, meaning that there exists $k^* \in [n]$, such that only the $k^*$ nearest neighbors of $x_0$ donate to the prediction of its label. Note that both $\alpha$ and $k^*$ may adapt from one $x_0$ to another. Also notice that the optimal weights depend on a single parameter $L/C$, namely the Lipschitz to noise ratio. As $L/C$ grows $k^*$ tends to be smaller, which is quite intuitive.

Without loss of generality, assume that the points are ordered in ascending order according to their distance from $x_0$, i.e., $d(x_1, x_0) \le d(x_2, x_0) \le \ldots \le d(x_n, x_0)$. Also, let $\boldsymbol{\beta} \in \mathbb{R}^n$ be such that $\beta_i = Ld(x_i, x_0)/C$. Then, the following is our main theorem:

**Theorem 3.1.** *There exists $\lambda > 0$ such that the optimal solution of (**P2**) is of the form*

$$\alpha_i^* = \frac{(\lambda - \beta_i) \cdot \mathbf{1}\{\beta_i < \lambda\}}{\sum_{i=1}^n (\lambda - \beta_i) \cdot \mathbf{1}\{\beta_i < \lambda\}}. \tag{1}$$

*Furthermore, the value of (**P2**) at the optimum is $C\lambda$.*

Following is a direct corollary of the above Theorem:

**Corollary 3.2.** *There exists $1 \le k^* \le n$ such that for the optimal solution of (**P2**) the following applies:*

$$\alpha_i^* > 0; \ \forall i \le k^* \quad and \quad \alpha_i^* = 0; \ \forall i > k^*.$$

*Proof of Theorem 3.1.* Notice that (**P2**) may be written as follows:

$$\min_{\boldsymbol{\alpha} \in \Delta_n} C\left(\|\boldsymbol{\alpha}\|_2 + \boldsymbol{\alpha}^\top \boldsymbol{\beta}\right) \qquad (\textbf{P2}).$$

We henceforth ignore the parameter $C$. In order to find the solution of (**P2**), let us first consider its Lagrangian:

$$L(\boldsymbol{\alpha}, \lambda, \boldsymbol{\theta}) = \|\boldsymbol{\alpha}\|_2 + \boldsymbol{\alpha}^\top \boldsymbol{\beta} + \lambda\left(1 - \sum_{i=1}^n \alpha_i\right) - \sum_{i=1}^n \theta_i \alpha_i,$$

where $\lambda \in \mathbb{R}$ is the multiplier of the equality constraint $\sum_i \alpha_i = 1$, and $\theta_1, \ldots, \theta_n \ge 0$ are the multipliers of the inequality constraints $\alpha_i \ge 0, \ \forall i \in [n]$. Since (**P2**) is convex, any solution satisfying the KKT conditions is a global minimum. Deriving the Lagrangian with respect to $\boldsymbol{\alpha}$, we get that for any $i \in [n]$:

$$\frac{\alpha_i}{\|\boldsymbol{\alpha}\|_2} = \lambda - \beta_i + \theta_i.$$

Denote by $\boldsymbol{\alpha}^*$ the optimal solution of (**P2**). By the KKT conditions, for any $\alpha_i^* > 0$ it follows that $\theta_i = 0$. Otherwise, for any $i$ such that $\alpha_i^* = 0$ it follows that $\theta_i \ge 0$, which implies $\lambda \le \beta_i$. Thus, for any nonzero weight $\alpha_i^* > 0$ the following holds:

$$\frac{\alpha_i^*}{\|\boldsymbol{\alpha}^*\|_2} = \lambda - \beta_i. \tag{2}$$

Squaring and summing Equation (2) over all the nonzero entries of $\boldsymbol{\alpha}$, we arrive at the following equation for $\lambda$:

$$1 = \sum_{\alpha_i^* > 0} \frac{(\alpha_i^*)^2}{\|\boldsymbol{\alpha}^*\|_2^2} = \sum_{\alpha_i^* > 0} (\lambda - \beta_i)^2. \tag{3}$$

**Algorithm 1** $k^*$-NN

---

**Input**: vector of ordered distances $\boldsymbol{\beta} \in \mathbb{R}^n$, noisy labels $y_1, \ldots, y_n \in \mathbb{R}$
Set: $\lambda_0 = \beta_1 + 1$, $k = 0$
**while** $\lambda_k > \beta_{k+1}$ and $k \leq n - 1$ **do**
   Update: $k \leftarrow k + 1$
   Calculate: $\lambda_k = \frac{1}{k} \left( \sum_{i=1}^k \beta_i + \sqrt{k + \left( \sum_{i=1}^k \beta_i \right)^2 - k \sum_{i=1}^k \beta_i^2} \right)$
**end while**
**Return**: estimation $\hat{f}(x_0) = \sum_i \alpha_i y_i$, where $\boldsymbol{\alpha} \in \Delta_n$ is a weight vector such $\alpha_i = \frac{(\lambda_k - \beta_i) \cdot \mathbf{1}\{\beta_i < \lambda_k\}}{\sum_{i=1}^n (\lambda_k - \beta_i) \cdot \mathbf{1}\{\beta_i < \lambda_k\}}$

---

Next, we show that the value of the objective at the optimum is $C\lambda$. Indeed, note that by Equation (2) and the equality constraint $\sum_i \alpha_i^* = 1$, any $\alpha_i^* > 0$ satisfies

$$\alpha_i^* = \frac{\lambda - \beta_i}{A}, \quad \text{where} \quad A = \sum_{\alpha_i^* > 0} (\lambda - \beta_i). \tag{4}$$

Plugging the above into the objective of (**P2**) yields

$$C \left( \|\boldsymbol{\alpha}^*\|_2 + \boldsymbol{\alpha}^{*\top} \boldsymbol{\beta} \right) = \frac{C}{A} \sqrt{\sum_{\alpha_i^* > 0} (\lambda - \beta_i)^2} + \frac{C}{A} \sum_{\alpha_i^* > 0} (\lambda - \beta_i)(\beta_i - \lambda + \lambda)$$

$$= \frac{C}{A} - \frac{C}{A} \sum_{\alpha_i^* > 0} (\lambda - \beta_i)^2 + \frac{C\lambda}{A} \sum_{\alpha_i^* > 0} (\lambda - \beta_i)$$

$$= C\lambda,$$

where in the last equality we used Equation (3), and substituted $A = \sum_{\alpha_i^* > 0} (\lambda - \beta_i)$. $\qquad\square$

## 3.1 Solving (**P2**) Efficiently

Note that (**P2**) is a convex optimization problem, and it can be therefore (*approximately*) solved efficiently, e.g., via any first order algorithm. Concretely, given an accuracy $\epsilon > 0$, any off-the-shelf convex optimization method would require a running time which is $\text{poly}(n, \frac{1}{\epsilon})$ in order to find an $\epsilon$-optimal solution to (**P2**)[2]. Note that the calculation of (the unsorted) $\boldsymbol{\beta}$ requires an additional computational cost of $O(nd)$.

Here we present an efficient method that computes the *exact* solution of (**P2**). In addition to the $O(nd)$ cost for calculating $\boldsymbol{\beta}$, our algorithm requires an $O(n \log n)$ cost for sorting the entries of $\boldsymbol{\beta}$, as well as an additional running time of $O(k^*)$, where $k^*$ is the number of non-zero elements at the optimum. Thus, the running time of our method is independent of any accuracy $\epsilon$, and may be significantly better compared to any off-the-shelf optimization method. Note that in some cases [26], using advanced data structures may decrease the cost of finding the nearest neighbors (i.e., the sorted $\boldsymbol{\beta}$), yielding a running time substantially smaller than $O(nd + n \log n)$.

Our method is depicted in Algorithm 1. Quite intuitively, the core idea is to greedily add neighbors according to their distance form $x_0$ until a stopping condition is fulfilled (indicating that we have found the optimal solution). Letting $\mathcal{C}_{\text{sortNN}}$, be the computational cost of calculating the sorted vector $\boldsymbol{\beta}$, the following theorem presents our guarantees.

**Theorem 3.3.** *Algorithm 1 finds the exact solution of* (**P2**) *within $k^*$ iterations, with an $O(k^* + \mathcal{C}_{sortNN})$ running time.*

*Proof of Theorem 3.3.* Denote by $\boldsymbol{\alpha}^*$ the optimal solution of (**P2**), and by $k^*$ the corresponding number of nonzero weights. By Corollary 3.2, these $k^*$ nonzero weights correspond to the $k^*$ smallest values of $\boldsymbol{\beta}$. Thus, we are left to show that (1) the optimal $\lambda$ is of the form calculated by the algorithm; and (2) the algorithm halts after exactly $k^*$ iterations and outputs the optimal solution.

Let us first find the optimal $\lambda$. Since the non-zero elements of the optimal solution correspond to the $k^*$ smallest values of $\boldsymbol{\beta}$, then Equation (3) is equivalent to the following quadratic equation in $\lambda$:

$$k^* \lambda^2 - 2\lambda \sum_{i=1}^{k^*} \beta_i + \left( \sum_{i=1}^{k^*} \beta_i^2 - 1 \right) = 0.$$

Solving for $\lambda$ and neglecting the solution that does not agree with $\alpha_i \geq 0, \ \forall i \in [n]$, we get

$$\lambda = \frac{1}{k^*} \left( \sum_{i=1}^{k^*} \beta_i + \sqrt{k^* + \left( \sum_{i=1}^{k^*} \beta_i \right)^2 - k^* \sum_{i=1}^{k^*} \beta_i^2} \right). \qquad (5)$$

The above implies that given $k^*$, the optimal solution (satisfying KKT) can be directly derived by a calculation of $\lambda$ according to Equation (5) and computing the $\alpha_i$'s according to Equation (1). Since Algorithm 1 calculates $\lambda$ and $\boldsymbol{\alpha}$ in the form appearing in Equations (5) and (1) respectively, it is therefore sufficient to show that it halts after exactly $k^*$ iterations in order to prove its optimality. The latter is a direct consequence of the following conditions:

    (1) Upon reaching iteration $k^*$ Algorithm 1 necessarily halts.

    (2) For any $k \leq k^*$ it holds that $\lambda_k \in \mathbb{R}$.

    (3) For any $k < k^*$ Algorithm 1 does not halt.

Note that the first condition together with the second condition imply that $\lambda_k$ is well defined until the algorithm halts (in the sense that the " $>$ "operation in the **while** condition is meaningful). The first condition together with the third condition imply that the algorithm halts after exactly $k^*$ iterations, which concludes the proof. We are now left to show that the above three conditions hold:

**Condition (1):** Note that upon reaching $k^*$, Algorithm 1 necessarily calculates the optimal $\lambda = \lambda_{k^*}$. Moreover, the entries of $\boldsymbol{\alpha}^*$ whose indices are greater than $k^*$ are necessarily zero, and in particular, $\alpha_{k^*+1}^* = 0$. By Equation (1), this implies that $\lambda_{k^*} \leq \beta_{k^*+1}$, and therefore the algorithm halts upon reaching $k^*$.

In order to establish conditions (2) and (3) we require the following lemma:

**Lemma 3.4.** *Let $\lambda_k$ be as calculated by Algorithm 1 at iteration $k$. Then, for any $k \leq k^*$ the following holds:*

$$\lambda_k = \min_{\boldsymbol{\alpha} \in \Delta_n^{(k)}} \left( \|\boldsymbol{\alpha}\|_2 + \boldsymbol{\alpha}^\top \boldsymbol{\beta} \right), \quad \text{where } \Delta_n^{(k)} = \{ \boldsymbol{\alpha} \in \Delta_n : \alpha_i = 0, \ \forall i > k \}$$

We are now ready to prove the remaining conditions.

**Condition (2):** Lemma 3.4 states that $\lambda_k$ is the solution of a convex program over a nonempty set, therefore $\lambda_k \in \mathbb{R}$.

**Condition (3):** By definition $\Delta_n^{(k)} \subset \Delta_n^{(k+1)}$ for any $k < n$. Therefore, Lemma 3.4 implies that $\lambda_k \geq \lambda_{k+1}$ for any $k < k^*$ (minimizing the same objective with stricter constraints yields a higher optimal value). Now assume by contradiction that Algorithm 1 halts at some $k_0 < k^*$, then the stopping condition of the algorithm implies that $\lambda_{k_0} \leq \beta_{k_0+1}$. Combining the latter with $\lambda_k \geq \lambda_{k+1}, \ \forall k \leq k^*$, and using $\beta_k \leq \beta_{k+1}, \ \forall k \leq n$, we conclude that:

$$\lambda_{k^*} \leq \lambda_{k_0+1} \leq \lambda_{k_0} \leq \beta_{k_0+1} \leq \beta_{k^*}.$$

The above implies that $\alpha_{k^*} = 0$ (see Equation (1)), which contradicts Corollary 3.2 and the definition of $k^*$.

**Running time:** Note that the main running time burden of Algorithm 1 is the calculation of $\lambda_k$ for any $k \leq k^*$. A naive calculation of $\lambda_k$ requires an $O(k)$ running time. However, note that $\lambda_k$ depends only on $\sum_{i=1}^{k} \beta_i$ and $\sum_{i=1}^{k} \beta_i^2$. Updating these sums incrementally implies that we require only $O(1)$ running time per iteration, yielding a total running time of $O(k^*)$. The remaining $O(\mathcal{C}_{\text{sortNN}})$ running time is required in order to calculate the (sorted) $\boldsymbol{\beta}$. □

## 3.2 Special Cases

The aim of this section is to discuss two special cases in which the bound of our algorithm coincides with familiar bounds in the literature, thus justifying the relaxed objective of (**P2**). We present here only a high-level description of both cases, and defer the formal details to the full version of the paper.

The solution of (**P2**) is a high probability upper-bound on the true prediction error $|\sum_{i=1}^{n} \alpha_i y_i - f(x_0)|$. Two interesting cases to consider in this context are $\beta_i = 0$ for all $i \in [n]$, and $\beta_1 = \ldots = \beta_n = \beta > 0$. In the first case, our algorithm includes all labels in the computation of $\lambda$, thus yielding a confidence bound of $2C\lambda = 2b\sqrt{(2/n)\log(2/\delta)}$ for the prediction error (with probability $1 - \delta$). Not surprisingly, this bound coincides with the standard Hoeffding bound for the task of estimating the mean value of a given distribution based on noisy observations drawn from this distribution. Since the latter is known to be tight (in general), so is the confidence bound obtained by our algorithm. In the second case as well, our algorithm will use all data points to arrive at the confidence bound $2C\lambda = 2Ld + 2b\sqrt{(2/n)\log(2/\delta)}$, where we denote $d(x_1, x_0) = \ldots = d(x_n, x_0) = d$. The second term is again tight by concentration arguments, whereas the first term cannot be improved due to Lipschitz property of $f(\cdot)$, thus yielding an overall tight confidence bound for our prediction in this case.

# 4 Experimental Results

The following experiments demonstrate the effectiveness of the proposed algorithm on several datasets. We start by presenting the baselines used for the comparison.

## 4.1 Baselines

**The standard k-NN:** Given $k$, the standard $k$-NN finds the $k$ nearest data points to $x_0$ (assume without loss of generality that these data points are $x_1, \ldots, x_k$), and then estimates $\hat{f}(x_0) = \frac{1}{k}\sum_{i=1}^{k} y_i$.

**The Nadaraya-Watson estimator:** This estimator assigns the data points with weights that are proportional to some given similarity kernel $K : \mathbb{R}^d \times \mathbb{R}^d \mapsto \mathbb{R}_+$. That is,

$$\hat{f}(x_0) = \frac{\sum_{i=1}^{n} K(x_i, x_0) y_i}{\sum_{i=1}^{n} K(x_i, x_0)}.$$

Popular choices of kernel functions include the Gaussian kernel $K(x_i, x_j) = \frac{1}{\sigma} e^{-\frac{\|x_i - x_j\|^2}{2\sigma^2}}$; Epanechnikov Kernel $K(x_i, x_j) = \frac{3}{4}\left(1 - \frac{\|x_i - x_j\|^2}{\sigma^2}\right) \mathbf{1}_{\{\|x_i - x_j\| \leq \sigma\}}$; and the triangular kernel $K(x_i, x_j) = \left(1 - \frac{\|x_i - x_j\|}{\sigma}\right) \mathbf{1}_{\{\|x_i - x_j\| \leq \sigma\}}$. Due to lack of space, we present here only the best performing kernel function among the three listed above (on the tested datasets), which is the Gaussian kernel.

## 4.2 Datasets

In our experiments we use 8 real-world datasets, all are available in the UCI repository website (`https://archive.ics.uci.edu/ml/`). In each of the datasets, the features vector consists of real values only, whereas the labels take different forms: in the first 6 datasets (QSAR, Diabetes, PopFailures, Sonar, Ionosphere, and Fertility), the labels are binary $y_i \in \{0, 1\}$. In the last two datasets (Slump and Yacht), the labels are real-valued. Note that our algorithm (as well as the other two baselines) applies to all datasets without requiring any adjustment. The number of samples $n$ and the dimension of each sample $d$ are given in Table 1 for each dataset.

| Dataset (n, d) | Standard k-NN | | Nadarays-Watson | | Our algorithm (k*-NN) | |
|---|---|---|---|---|---|---|
| | Error (STD) | Value of k | Error (STD) | Value of $\sigma$ | Error (STD) | Range of k |
| QSAR (1055,41) | 0.2467 (0.3445) | 2 | 0.2303 (0.3500) | 0.1 | **0.2105* (0.3935)** | 1-4 |
| Diabetes (1151,19) | 0.3809 (0.2939) | 4 | 0.3675 (0.3983) | 0.1 | **0.3666 (0.3897)** | 1-9 |
| PopFailures (360,18) | 0.1333 (0.2924) | 2 | **0.1155 (0.2900)** | 0.01 | 0.1218 (0.2302) | 2-24 |
| Sonar (208,60) | 0.1731 (0.3801) | 1 | 0.1711 (0.3747) | 0.1 | **0.1636 (0.3661)** | 1-2 |
| Ionosphere (351,34) | 0.1257 (0.3055) | 2 | 0.1191 (0.2937) | 0.5 | **0.1113* (0.3008)** | 1-4 |
| Fertility (100,9) | 0.1900 (0.3881) | 1 | 0.1884 (0.3787) | 0.1 | **0.1760 (0.3094)** | 1-5 |
| Slump (103,9) | 3.4944 (3.3042) | 4 | 2.9154 (2.8930) | 0.05 | **2.8057 (2.7886)** | 1-4 |
| Yacht (308,6) | 6.4643 (10.2463) | 2 | 5.2577 (8.7051) | 0.05 | **5.0418* (8.6502)** | 1-3 |

Table 1: Experimental results. The values of $k$, $\sigma$ and $L/C$ are determined via 5-fold cross validation on the validation set. These value are then used on the test set to generate the (absolute) error rates presented in the table. In each line, the best result is marked with bold font, where asterisk indicates significance level of $0.05$ over the second best result.

### 4.3 Experimental Setup

We randomly divide each dataset into two halves (one used for validation and the other for test). On the first half (the validation set), we run the two baselines and our algorithm with different values of $k$, $\sigma$ and $L/C$ (respectively), using 5-fold cross validation. Specifically, we consider values of $k$ in $\{1, 2, \ldots, 10\}$ and values of $\sigma$ and $L/C$ in $\{0.001, 0.005, 0.01, 0.05, 0.1, 0.5, 1, 5, 10\}$. The best values of $k$, $\sigma$ and $L/C$ are then used in the second half of the dataset (the test set) to obtain the results presented in Table 1. For our algorithm, the range of $k$ that corresponds to the selection of $L/C$ is also given. Notice that we present here the average absolute error of our prediction, as a consequence of our theoretical guarantees.

### 4.4 Results and Discussion

As evidenced by Table 1, our algorithm outperforms the baselines on 7 (out of 8) datasets, where on 3 datasets the outperformance is significant. It can also be seen that whereas the standard $k$-NN is restricted to choose one value of $k$ per dataset, our algorithm fully utilizes the ability to choose $k$ adaptively per data point. This validates our theoretical findings, and highlights the advantage of adaptive selection of $k$.

## 5 Conclusions and Future Directions

We have introduced a principled approach to locally weighted optimal estimation. By explicitly phrasing the bias-variance tradeoff, we defined the notion of optimal weights and optimal number of neighbors per decision point, and consequently devised an efficient method to extract them. Note that our approach could be extended to handle multiclass classification, as well as scenarios in which predictions of different data points correlate (and we have an estimate of their correlations). Due to lack of space we leave these extensions to the full version of the paper.

A shortcoming of current non-parametric methods, including our $k^*$-NN algorithm, is their limited geometrical perspective. Concretely, all of these methods only consider the distances between the decision point and dataset points, i.e., $\{d(x_0, x_i)\}_{i=1}^n$, and *ignore* the geometrical relation between the dataset points, i.e., $\{d(x_i, x_j)\}_{i,j=1}^n$. We believe that our approach opens an avenue for taking advantage of this additional geometrical information, which may have a great affect over the quality of our predictions.

## Footnotes

[1]Note that our analysis holds for both setups of classification/regression. For brevity we use a *classification* task terminology, relating to the $y_i$'s as *labels*. Our analysis extends directly to the regression setup.

[2]Note that (**P2**) is not strongly-convex, and therefore the polynomial dependence on $1/\epsilon$ rather than $\log(1/\epsilon)$ for first order methods. Other methods such as the Ellipsoid depend logarithmically on $1/\epsilon$, but suffer a worse dependence on $n$ compared to first order methods.

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
