[Reviews · NeurIPS 2016]

Reviewer 1

Summary

The paper investigates the $k$-nearest neighbors algorithm. In a setting of the form $y=f(x)+\epsilon$, the authors introduce their work by the following quantity to be minimized: $\abs{\sum_i \alpha_i y_i - f(x_0)}$ over $\alpha$ in the $\mathbb{R}^n$-simplex. The authors then relax this problem to a problem (P2) and provide an algorithm to efficiently solve (P2). The algorithmic cost of this solver is given, and simulations on eight UCI datasets, on which the proposed predictor significantly outperforms the Nadaraya-Watson estimator and $k-$NN in three of them.

Qualitative Assessment

The paper investigates an interesting problem and provides an original approach. I mostly have concerns about the clarity of the proposed manuscript. * The authors advocate their work offers a unified setting to address both classification and regression (footnote, page 3). I find that statement misleading since the paper alternates regression and classification notation and vocabulary. I feel the authors should rephrase some parts and avoid ambiguity whevener possible. * My most serious concern is about the general message: in my initial readings, I believed solving the surrogate problem (P2) would yield generalisation bounds on the $k$NN algorithm. I urge the authors to prevent any misleading by stating that solving (P2) incidently yields an absolute upper bound on the optimum of (P1), yet this serves as an introduction to the actual problem addressed by the paper (P2) and its efficient solving (Algorithm 1, supported by a simulation study). * I would find interesting a short discussion on the choice of the L1 norm to assess the performance of the algorithm. If possible, a comparison with L0 (in classification) and/or L2 (regression) norms in the simulation would help decide wether the proposed algorithm significantly outperforms standard $k$NN. * Lines 28-30, sentence "Given a new... the ground truth).". As written above this is not quite what is achieved in the paper and this might mislead readers as to what the actual contributions of the paper are. * Minor points and typos: - Line 11 and Table 1: Nadarays -> Nadaraya - Line 54: choosing -> choose - Lines 236, 243 and 264: please correct to proper citations, where all the authors are credited (no "et al.") - Line 264: capitalize Nadaraya-Watson

Confidence in this Review

1-Less confident (might not have understood significant parts)


Reviewer 2

Summary

The autors propose a new kNN algorithm where the prediction task is cast as a regularized optimization. The regularization framework, combined with the choice of the loss, yields an algorithm that automatically select the number of neighbors and their weight in the voting process. An algorithmic procedure is provided to solve the regularized optimization, and the procedure is illustrated on classical datasets.

Qualitative Assessment

The authors consider a L1 loss rather than the classical L2 (for regression) or Hard-Loss (for classif). While I can see the purpose of this choice from a technical point of view, in the classification framework the relationship between the classification performance (ie Hard Loss) and the performance as quantified by the loss suggested by the authors is not straightforward. It could have been of interest to report the classical classification error rate along with the average absolute error in the experimental results to check whether the proposed procedure significantly improves the classification performance or not. In section 3 the guarantee regarding the computational cost does not seem really attractive: basically the authors propose to fully order the training samples according to their distance to the point to be classified, which results in a complexity of nlog(n). Since the procedure they propose makes use on the k^* nearest neighbors only, it seems that classical efficient methods such as kdtree could be adapted to efficiently find the relevant neighbors. If not, can the authors point out the reason why this would be a problem ? In practice the authors could have provided in Suppl.Mat. the computational time associated with each method they tried. In section 4.3 it is written that the algorithm "fully utilizes the ability to choose k adaptively per point". It is true that the method locally chooses the number of points and the weights. Still the upper-bounding of the loss in p3 discards the local characteristics of the regression function through the global Lipschitz property, which can be locally quite loose. As a consequence, for parameter k to be at least 2 for instance, it requires $\beta_2$ to be higher than $\beta_1+1$, whether locally the regression function is close to 0.5 (in the classification case) or to 0 or 1. There may be room for improvement here in terms of fully exploiting the ability to choose k locally.

Confidence in this Review

2-Confident (read it all; understood it all reasonably well)


Reviewer 3

Summary

This paper proposes a reweighting method of nearest neighbors (NNs) for NN classification and regression. The authors derive the upper bound of the difference between the model output and the target value, and the minimization of the upper bound is obtained by a simple and quick iterative process. As a result, the weights \alpha for k nearest neighbors are determined for given data, as well as the optimal number k.

Qualitative Assessment

The mathematical derivation in this work is novel and interesting, but the writing can be improved. The derivation uses Hoeffding’s inequality for noise on labels, with a mild assumption on the maximum noise. The analysis using KKT condition is straightforward, and it results in an interesting strategy for finding weights. I want to support the acceptance of this paper, but I have several concerns: 1) According to the derivation, the upper bound of the risk (P1) is given as C\lambda (line 112). With some simple calculations using appropriate parameters, this upper bound does not give a small value. For example, with b=1, delta=0.1, and k*=10, and \lambda greater than the minimum (\beta_i + 1/k*) from Eq.(4), the minimum upper bound I roughly calculated is about 0.24. For example in classification, this does not seem to be a meaningful upper bound. Can I reduce the upper bound further with appropriate k* (k*=3-10)? 2) In high dimensional space, distance to the nearest neighbors is unreliable in many cases. In particular, the distance is highly concentrated, and the nearest neighbor does not appear very closely to the test point. How do the authors advocate the Lipschitz condition that the function deviation is bounded linearly by the distance within the range where the nearest neighbors appear? 3) Though the derivation and the result is interesting, the paper is not well-written. The structure and the writing can be improved. 4) In the discussion from line 170 to 179, the two presented examples are redundant. The first example is the special case of the second example. 5) The title has to be changed. From the title, I cannot have any information about this paper. The derivation in this manuscript is new, and I like the way of obtaining the proposed optimization procedure. However, further honest analysis will make this paper more interesting when this method is useful and when it is not. Also, if the authors can provide some guarantee that the writing can be significantly improved, I want to recommend the acceptance of this paper.

Confidence in this Review

3-Expert (read the paper in detail, know the area, quite certain of my opinion)


Reviewer 4

Summary

The paper considers the problem of weighted k-nearest neighbors for regression and classification. They propose a method to choose k and the weights at a given data point x; under an additive statistical model, it minimizes the MSE of the prediction. Through some theoretical derivations, they arrive at an algorithm that has cost approximately linear in the number of observations. Experiments on real-world data show that in many data sets the proposed algorithm outperforms standard k-nearest neighbors and the Nadaraya-Watson estimator.

Qualitative Assessment

Clarity and presentation: The paper is well-written and organized and the authors clearly communicate their ideas. I like the structure of the proof of Theorem 3.3. However, please proof-read the paper for a few grammatical errors and typos. I would also suggest shortening the proofs given in the paper, possibly pushing the full versions to the supplement. The paper seems a bit long and doing so would also give more room to expand on the experiments. Potential impact and usefulness: The proposed method is able to combine the choice of k and the weights, which is great, and adapt prediction to each data point, which could only improve results. My only concern is that it may be too computationally expensive when we want to obtain a decision boundary (or predict at many points) or when the data size is large. Novelty: Although the idea of adapting prediction to each data point x by minimizing the MSE is not new, I have not seen it applied to k-nearest neighbors regression and classification. Technical Quality: The proofs seem to be complete and sound, the level of rigor is good, and the experimental methods are appropriate. The problem definition was done well, and I liked how the authors drew connections to other bounds in the literature. I do have several questions/concerns: 1. How do we compute L/C (and thus beta) when f is unknown? This is mentioned briefly in the experiments, but I believe this is a very important issue. 2. In many cases, we want a decision boundary, or at least predictions at many different values of x. Is there an efficient way to get it? 3. In the experiments, you should show computation times of the various algorithms on the data sets, because that is important for k-nearest neighbors in practice. Also, it would be helpful to describe the data sets you use further. For example, do they exhibit the behavior shown in Figure 1?; what size are they?

Confidence in this Review

2-Confident (read it all; understood it all reasonably well)


Reviewer 5

Summary

This paper shows that a probabilistic upper bound of absolute error given the input point can be defined as the problem (P2), and proposes a solver for this problem as in Algorithm 1. The upper bound is derived through Lipschitz continuity and the Hoeffding's inequality. There's one hyperparameter "L/C" appearing in the upper bound that is a function of three unknown quantity, Lipschitz continuity of the true function L, a bound on the data set noise b, and probability of Hoeffding's inequality violation delta. The optimization algorithm is derived by casting the coefficient optimization to a threshold (lambda) optimization problem. Leveraging the increasing property of lambda, the paper showed that the greedy approach (Algorithm 1) can achieve the optimal.

Qualitative Assessment

This paper is clearly written, technically sound, and novel. Only concern for me is the practical usefulness of exact k-NN. It will be better if the authors can mention about compatibility with some approximative k-NN methods. Other minor points: 1. Showing the optimal L/C in the experimental section will help understanding relation between "range of k" and L/C. I'm also interested in average k of each task. 2. Indicator function in Theorem 3.1 should be defined for clarity.

Confidence in this Review

2-Confident (read it all; understood it all reasonably well)